# Comparative transcriptomics of leaves of five mulberry accessions and cataloguing structural and expression variants for future prospects

Vinay Kumar Baranwal[1,2☯], Nisha Negi[1☯], Paramjit Khurana[1]*

**1** Department of Plant Molecular Biology, University of Delhi South Campus, New Delhi, India, **2** Department of Botany, Swami Devanand Post Graduate College, Math-Lar, Lar, Deoria, Uttar Pradesh, India

☯ These authors contributed equally to this work.
* param@genomeindia.org

**Data Availability Statement:** The sequence data for leaf transcriptome of Morus laevigata and Morus serrata is already submitted to NCBI SRA database with accession numbers SRP068061 and

## Abstract

*Bombyx* mori, a monophagous insect, prefers leaves of the certain species of *Morus* more than others. The preference has been attributed to morphological and anatomical features and biochemical compounds. In the present manuscript a comparison has been made among the transcriptome of leaves of the two preferred cultivated varieties and three wild types species. While assembling, high quality transcriptomes of five genotypes were constructed with a total of 100930, 151245, 89724, 181761 and 102908 transcripts from ML, MN, MS, K2 and V1 samples respectively. Further, to compare them, orthologs were identified from these assembled transcriptome. A total of 22462, 23413, 23685, 24371, 18362, 22326, 20058, 18049, 17567 and 20518 clusters of orthologs were found in one to one comparison in KvsN, KvsL, KvsS, KvsV, LvsN, LvsS, LvsV, NvsS, NvsV, and SvsV respectively. 4236 orthologs with algebraic connectivity of 1.0 were then used to compare and to find out differentially expressed transcripts from all the genotypes. A total of 1037 transcripts expressed that include some of the important morphology, anatomy and biochemical pathways regulating transcription factors (AP2/ERFs and C2H2 Zinc fingers) and signalling components were identified to express differentially. Further, these transcriptomes were used find out markers (SSR) and variants and a total of 1101013, 537245, 970877, 310437, 675772, 338400, 581189, 751477, 514999 and 257107 variants including SNP, MNP, Insertions and deletions were found in one to one comparisons. Taken together, our data could be highly useful for mulberry community worldwide as it could be utilized in mulberry breeding programs.

## Introduction

*Morus* is the only tree species on which the monophagous silk worms thrives. This unique association has been historically utilized by humans. This has traditionally contributed to the

SRP067869 respectively by our lab previously. The read data for Morus notabilis leaf transcriptome was downloaded from NCBI SRA database with accession number SRA075563. The sequence data for Morus indica cv K2 and cv V1 could be accessed under the Bioprojects accesions PRJNA717975 and PRJNA717991 respectively.

**Funding:** PK is thankful to the Department of Biotechnology, Ministry of Science and Technology, Government of India for providing grants for the current research work with Grant Number BT/PR13457/PBD/19/214/2010. The funders had no role in study design, data collection and analysis, decision to publish, or preparation of the manuscript.

**Competing interests:** The authors have declared that no competing interests exist.

humankind in form of its second most important requirements of clothing after food. The origin of *Morus* is predicted to be the high latitude regions in Laurasia [1]. The mountains between 20–40˚N latitudes in East Asia are the secondary origin centres of *Morus* [1]. Spread of *Morus* from its centre of primary and secondary origins to the other part of the world has been attributed to faster evolutionary rates [2]. The biological diversity exhibited by *Morus* is also a result of the faster evolution as well as local adaptation. This variation could be utilized in form of primary gene pool for *Morus* improvement programmes.

The most important part of the *Morus* plant from the view point of silk-worm consumption is its leaf. Leaves are the sole source of nutrition for monophagous silk-worm. Its morphological, physiological, biochemical and molecular features have direct bearing on its consumption by the silk-worm and ultimately the yield of silk from it [3]. Absence of trichomes, tracheids, flaccidness and high water retention capability, high protein and carbohydrate content, presence or absence of certain phenolics have been shown to be some traits preferred by silk-worm [4]. Further, *Morus* leaves become non palatable due to its susceptibility to various kind of stresses including abiotic and biotic stresses, numerous biological and physiological diseases [5]. These conditions cause a hefty toll on the yield of leaves and ultimately on silk production globally. Further, it has been found that feeding of silkworm on different variety also modulate the quality and quantity of the silk produced. Among the four varieties including Ichinose, Kenmochi, Kines, and local, when insects were feed on Kines, they performed better [6]. In another bromatological study comprising mulberry cultivars Miura (standard), Korin and Tailandesa and four hybrids FM 3/3, FM 86, SK 1 and SK 4, SK4 was found to be the best source of nutrition for insects [7]. Most of the sericulture researches are focused towards improving the quality and yield of leaves in burgeoning hostile conditions. *Morus* has also found its uses in dietary, medicinal, timber, recreation etc. Identification of the important genes possessed by different genotypes and their integration into improved varieties through traditional breeding approaches is time consuming.

India is bestowed upon with numerous wild varieties [8] of *Morus* and some of the varieties are created by selection methods as well as breeding efforts to cater to the need of sericulture [9]. Two of the wild species namely *Morus laevigata* (ML) and *Morus serrata* (MS) have been claimed to harbour beneficial traits [4] and are considered as good parenting stocks for hybrid creation, gene transfer and other such usages [10]. Further, recently sequenced *Morus notabilis* (MN) [2] is also a wild species growing mainly in Sichuan province of the Peoples Republic of China that harbours array of genes which could be effectively utilized to increase the yield of leaves of other *Morus* species. Among the cultivated varieties of India, K2 (Kanva 2) [11] and V1 (Victory 1) are well known for their improvised uses. K2 was developed at CSRTI, Mysore, India belongs to *Morus indica* and was selected from natural seedling population of Mysore local variety suitable for rain fed and irrigated regions of Southern India. Similarly, Victory-1 (V1) was also developed at CSRTI from a cross of two indica accessions S-30 × C776. V1 was recommended its cultivation in irrigated condition in South Indian states [12]. A brief account of these genotypes is presented below. Texture of MS leaf varies from thin to thick, chartaceous to coriacious, tomentose and velvety and from unlobed to multilobed in some local accession. MS leaves are coarse and thicker, hairy leaves with greater moisture retention capacity [8, 13]. Leaves of ML are mostly unlobed, rough and thick. Further, they are pale green in color and slightly coarse in texture [14]. These properties render these wild species seldom usable for rearing silk worms. Contrary to this, cultivated varieties of *Morus indica* (MI) i.e. K2 and V1 have promoting leaf features. K2 leaves are unlobed, ovate or broadly ovate and smooth in texture [12]. Similarly, V1 leaves are thick, glossy, ovate, smooth and dark green in color [12]. Further, protein and carbohydrate content in these two varieties were found to 21% and 11.5% in K2 and 24.6% and 16.98% in V1 respectively [15, 16].

Despite these advantageous features, cultivated varieties have known demerits which restrict their usefulness in hostile conditions and require improvements. K2 and V1 are reported to be drought susceptible and are not recommended for cultivation in drought vulnerable areas [17]. Further, K2 is susceptible to leaf roller pest caused by *Diaphaniapulverulentalis* which causes reduction in leaf yield [18]. V1 is moderately resistant to leaf rust causal organism *Cercospora moricola* and tukra disease caused by pink mealy bug *Maconellicoccus hirsutus* [19]. On the other hand, wild varieties like ML possess genes for termite resistance [5, 20], powdery mildew [21, 22] and leaf spot resistance [4, 23]. Some accessions of ML collected from coastal regions of Andaman showed salt stress tolerance with heritability of this trait [10]. MS is known for its tolerance to cold and drought conditions [20, 21].

A comparative account of transcriptome of the leaves of five genotypes including MS, ML, MN, K2 and V1 is presented in this manuscript. This study was performed to understand the similarities and differences in these genotypes at molecular level.

With the advent of improved functional genomics tools like next generation sequencing etc, it could be effectively utilized to study differences of transcriptome which could be used to identify genes responsible for unique behaviour of the genotype in question [24]. Furthermore, the same dataset could be used to identify variants which are in close association with the identified genes and could hasten the improvement program [23]. Thus with these objectives in our mind we have sequenced transcriptome of four Indian genotypes and included the data of already sequenced one Chinese wild type to find out important genes contributing to beneficial traits and markers for future uses.

## Results

### Assembly of *Morus* transcriptomes

In the present manuscript, transcriptome data of four species of *Morus* including ML, MS, MI and MN have been analyzed. For MI two cultivars namely K2 and V1 were taken. An analysis was conducted with ML samples (due to maximum number of reads from single sample) with different assemblers including Tansabyss (version 2.0.1) [25], CLC Genomics Workbench (6.9.1) (Qiagen Gmbh; Germany), SOAPdenovo (version 2.04) [26] and Trinity (versionr21040413p1) [27] in order to find out the best suited assembler for the studied samples. N50 values of 1745, 232, 605 and 592 were obtained from Trinity, SOAPdenovo, TransABByss and CLC Genomics Workbench respectively. Further, average sequence length of 914.54, 181.65, 505.91 and 532.2 and maximum sequence length of 20750, 11583, 24279 and 14719 base pairs were reported by Trinity, SOAPdenovo, TransABByss and CLC Genomics workbench respectively (S1 Table). This analysis suggests that Trinity has performed better with respect to other assemblers used in this study. Hence, Trinity assembler was used for all the samples used in this study to construct their respective transcriptomes which were used further in downstream analyses. When Trinity was used as a default assembler for all five samples, a total of 100930, 151245, 89724, 181761 and 102908 transcripts were assembled from ML, MN, MS, K2 and V1 samples respectively. Further, N50 values these samples were found to be equal to 1728, 943, 1810, 1745 and 1882 from ML, MN, MS, K2 and V1 samples respectively suggesting a good assembled transcriptome for downstream analyses. Table 1 summarize different statistics obtained from the Trinity assembled transcriptomes of all the five samples considered in this study.

### Annotation and orthologs identification from different species

To annotate the assembled sequences, Annocript [28] bundled scripts and software were used which rely on BLAST tool,UniRef (version 2014_04) coupled with SwissProt databases

**Table 1. Showing the different stats of Trinity assemblies of different species and two cultivars.**

|  | *M laevigata* | *M notabilis* | *M serrata* | K2 | V1 |
|---|---|---|---|---|---|
| Total sequences | 100930 | 151245 | 89724 | 181761 | 102908 |
| Total bases | 95915947 | 92057999 | 87504074 | 166227130 | 105415355 |
| Min sequence length | 201 | 201 | 201 | 201 | 201 |
| Max sequence length | 14763 | 12307 | 16769 | 20750 | 22648 |
| Average sequence length | 950.32 | 608.67 | 975.26 | 914.54 | 1024.37 |
| Median sequence length | 503 | 335 | 502 | 446 | 518.5 |
| N25 length | 2766 | 1916 | 2867 | 2863 | 3036 |
| N50 length | 1728 | 943 | 1810 | 1745 | 1882 |
| N75 length | 791 | 374 | 834 | 748 | 935 |
| N90 length | 358 | 259 | 360 | 326 | 373 |
| N95 length | 270 | 230 | 270 | 257 | 273 |
| As | 29.55% | 27.29% | 29.55% | 29.06% | 29.47% |
| Ts | 29.56% | 27.20% | 29.44% | 29.02% | 29.33% |
| Gs | 20.60% | 23.07% | 20.66% | 21.09% | 20.67% |
| Cs | 20.28% | 22.45% | 20.34% | 20.83% | 20.54% |
| (A + T)s | 59.11% | 54.49% | 59.00% | 58.08% | 58.79% |
| (G + C)s | 40.89% | 45.51% | 41.00% | 41.92% | 41.21% |
| Ns | 0.00% | 0.00% | 0.00% | 0.00% | 0.00% |

(version 2014_04), Rfam (version 12.1), Conserved Domains Database (CDD; version 3.14) of NCBI and Pfam database (version 27.0). It uses a similarity search approach and by utilizing the above mentioned databases, allocates annotations including conserved domain, Pfam domain, KEGG pathways and GO terms. All five assemblies generated using Trinity assembler were annotated using the Annocript pipeline and were used in downstream analysis. In order to identify orthologs from different genotypes, Proteinortho program (version 5.16) [29] was used with default parameters (including e value < = 1e-05, minimum percent identity = 25 and minimum coverage percent = 50). Proteinortho is a program which could be installed on multiple processor system with distributed architecture. Further, it is designed to use large datasets such as assembled transcriptome or the predicted proteomes by implementing an extended version of the reciprocal best alignment methods. It results in identification of clusters with higher algebraic connectivity value based on which clusters of orthologs two or more genotypes could be identified. In this analysis the program was used with default values and 22462, 23413, 23685, 24371, 18362, 22326, 20058, 18049, 17567 and 20518 clusters were found in one to one comparison in KvsN, KvsL, KvsS, KvsV, LvsN, LvsS, LvsV, NvsS, NvsV, and SvsV respectively. The numbers of transcripts in these clusters from different genotypes are given (Fig 1). Further, in downstream analyses like differential expression analysis, identification of In-dels etc, only those transcripts were used which formed a statistically significant cluster with default cutoff of algebraic connectivity ($\geq$0.85).

## Cross species leaf samples differential expression analysis and validation of genes using Real-time PCR

For avoiding noise in real time pcr data, a stringent criterion of algebraic connectivity of 1.0 was applied to all the orthologs. The orthologs showing algebraic connectivity of 1.0 were selected and expression analyses were conducted across the leaves stages of all the genotypes. A cluster of 4236 genes was used where every transcript has a corresponding gene and single transcript was predicted in all five species in this study. When FDR p value 0.001 and a fold

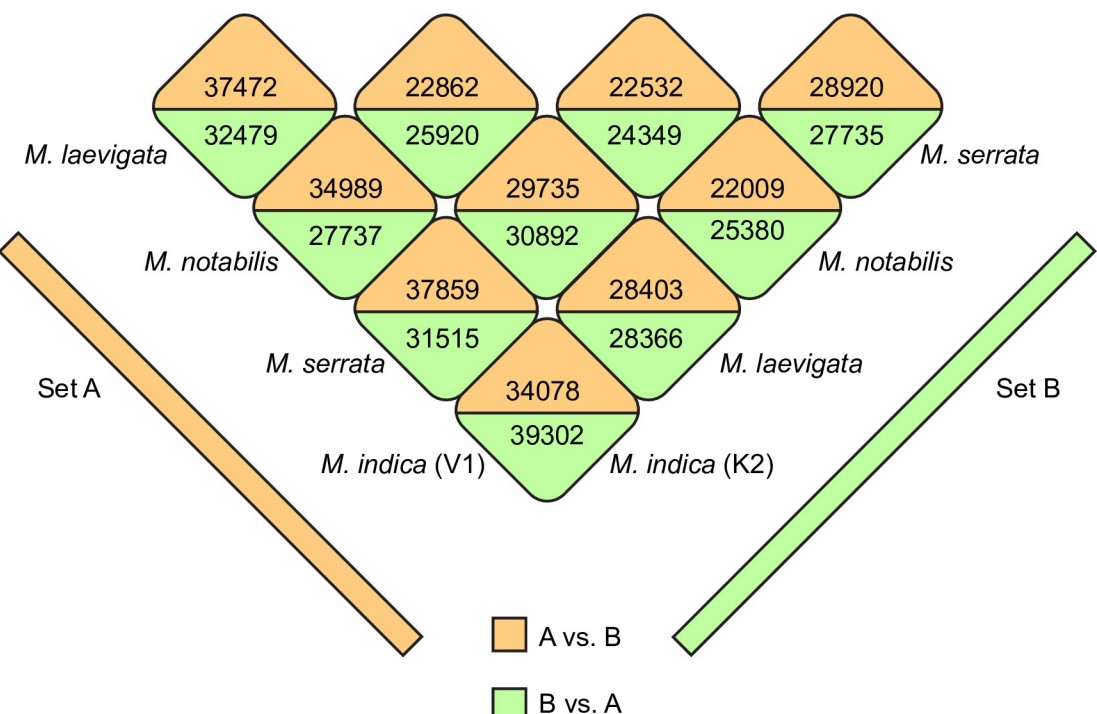

**Fig 1. Figure showing the number of orthologs reported in assembled transcriptomes from different genotypes using Proteinortho.** Trinity assembled transcriptome was used with default parameters. Numbers represent the transcripts in one to one comparison when reference transcriptome was used as query as well as source. See legend for details.

change $\geq$ 2.0 was applied, out of 4236 genes, a total of 1037 genes were found to express differentially. On increasing the stringency of fold change $\geq$ 10, analysis showed that a total of 91 genes were found to express differentially across the leaf samples of different genotypes which are shown in heat map (Fig 2).

On the basis of the contrasting expression profiles exhibited by the *in silico* expression analysis, 9 genes were selected (S2 Table) for the validation of transcriptome data by qPCR. qPCR also exhibited similar pattern obtained by transcriptome analysis. C _8555 (Morus018966) annotated as Cytochrome P450 71A1 in morusDB database and Genbank showed down-regulation in ML and MS. C_24691(Morus001314) annotated as AP2-like ethylene-responsive transcription factor AIL1 showed significant up-regulation in ML and V1 while it was down-regulated in MS and K2. c27976 (Morus014999) annotated as transcription factor bHLH162 showed five-fold up-regulation in MS with fold change of three in ML and much lower in V1 and K2. C29817 (Morus027832) annotated as UDP-glycosyltransferase 74F2 was found to be up-regulated in ML and V1, with highest fold change of nearly 2.5 fold in ML followed by V1 and K2 with almost similar expression level in the two cultivated varieties, while its expression was negligible in MS. c30253 (Morus020805) which is a putative ubiquitin-conjugating enzyme E2 showed significantly high level of expression in ML of about 8 fold as compared to K2. This gene showed almost ~3.0 fold up-regulation in V1, however almost similar expression level was reported in MS with respect to K2. C_30795 (Morus002089) annotated as putative S-acyltransferase showed a mild up-regulation of ~1.75 fold and ~2.25 fold in V1 and ML respectively with respect to K2. This gene did not exhibit any significant difference in MS with respect to K2. Transcripts of another gene i.e. C_31586 (Morus003231) which is putative aquaporin PIP1-4 showed down-regulation in almost all the genotypes with respect to K2.

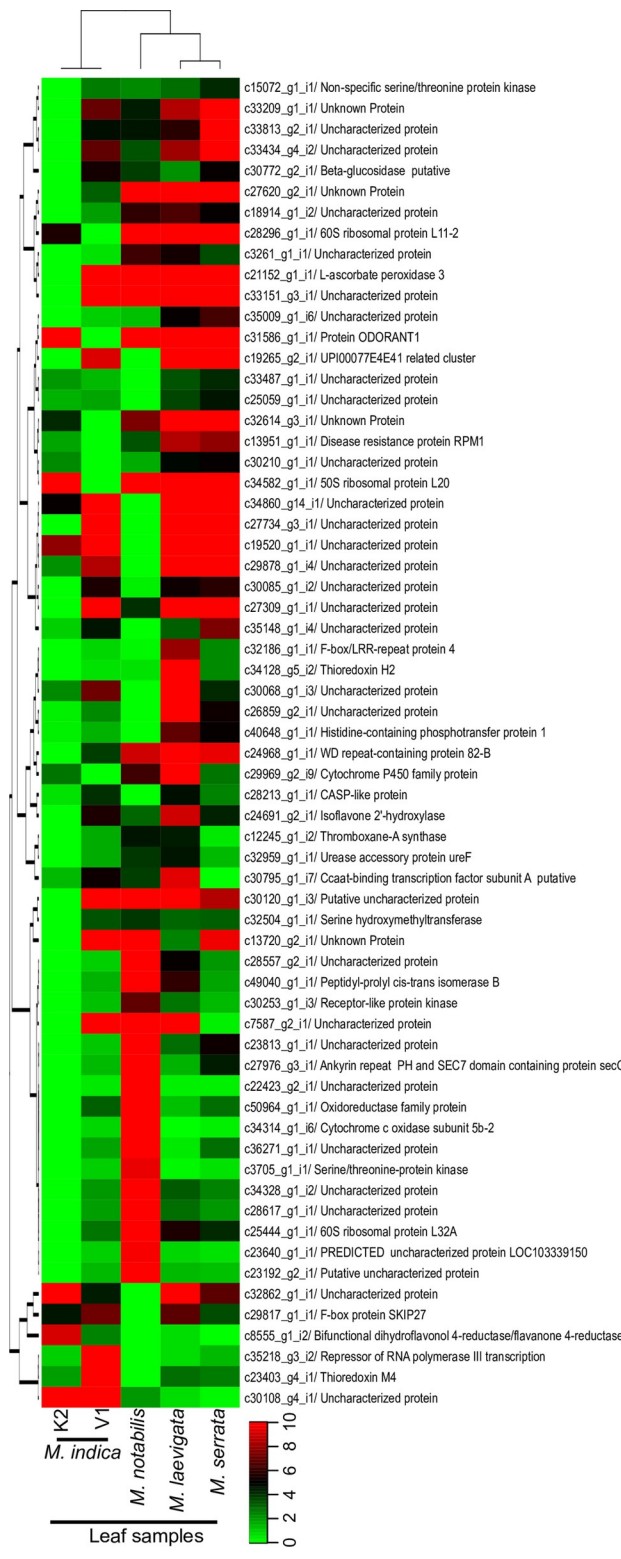

**Fig 2. Heatmap showing the relative expression pattern exhibited by the $> = 10$ fold change showing differentially expressed genes in five leaves samples of three genotypes and two cultivars from a fourth species.** Mean normalized FPKM values obtained were used to plot the heatmap after applying Ward's correction and Manhattan clustering.

However, this down-regulation was not significant enough in ML. C_32186 (Morus024808) Short-chain dehydrogenase/reductase family 42E member 1 showed significantly higher expression of all in ML leaves compared to K2. The expression of this gene was almost similar in rest two genotypes i.e. V1 and MS compared to K2. C_40648 (Morus013816) Zinc finger CCCH domain-containing protein showed higher expression in three genotypes i.e. V1, ML and MS and the fold change was highest in ML followed by V1 and MS (Fig 3). Pearson correlation coefficient was also calculated for normalized RNA-seq expression fold change values (TPM values) and the fold change values obtained by qPCR experiments (S3 Table). Out of nine selected genes for qPCR, c31586 showed negative correlation between qPCR and RNA-seq (-0.145), and rest nine showed positive correlations. The least correlation between RNA-seq and qPCR was observedfor c8555 (0.5926). For rest of the genes, the correlation was in the range of >0.7. This further confirms the robustness of the RNA-seq data.

## Identification of variants across genotypes and their effect prediction

Transcriptome data has been exploited to identify variants from non model system especially in plants. Identification of several different types of variants from assembled transcriptome has been done from many crop plants. Further, the variants identified from these datasets are from the expressed region of the genome, there is a higher probability of being functional. Furthermore, almost all different types of variants including SSRs, SNPs, Insertions, and Deletions etc. could be identified. Algorithms have been developed to identify haplotypes and their effect could be predicted. Variants have been identified using Freebayes and their effect have been predicted using SNPeff from all possible ten one to one comparisons namely K2vsN, K2vsL, K2vsS, K2vsV, LvsN, LvsS, LvsV, NvsS, NvsV, and SvsV respectively. A total of 1101013, 537245, 970877, 310437, 675772, 338400, 581189, 751477, 514999 and 257107 variants including SNP, MNP, Insertions and deletions were found in K2vsL, K2vsS, K2vsN, K2vsV1, LvsN, LvsV1, LvsS, NvsS, NvsV1 and SvsV1 comparisons respectively (Fig 4A). Of these variant majority were found of SNP type with 978812, 472657, 852388, 278564, 578099, 297490, 501129, 627978, 441704 and 226173 SNPs in K2vsL, K2vsS, K2vsN, K2vsV1, LvsN, LvsV1, LvsS, NvsS, NvsV1 and SvsV1 comparisons respectively. Variants identified would have effects on transcript ranging from null to severe. In order to study the impacts of identified variants SNPeff was used. In the pair wise comparison K2vsL, a total of 297701 variants were identified which could be split into 138674, 1068 and 157959 variants in miss-sense, non-sense and silent mutations categories respectively (Fig 4B). Similarly, 139527, 303324, 86324, 207580, 96981, 149284, 210513, 161169, 70464 total variants from these categories were identified from K2vsL, K2vsS, K2vsN, K2vsV1, LvsN, LvsV1, LvsS, NvsS, NvsV1 and SvsV1 comparisons respectively. In all the pair wise comparison, the silent mutations were higher with respect to non-sense as well as silent except for LvsN where non-sense mutations were reported to be highest. In the SNPs transition to transversion ratios were found minimum in LvsN (1.63) and maximum in K2vsL (1.74) (Fig 4C). Further, the variants were analyzed with their custom generated GFF annotation files in order to identify the genomic location in which they fall and how does it impact the overall transcriptome of all the species studied. This way all the variants could be categorized into high, low, moderate and modifier categories which have gradually lesser impacts due to identified variants. Majority of these variants find their place in modifier category with highest in K2vsS comparison (73.35%) and lowest in NvsV1 (65.82%). Variants that have high impact were relatively minuscule in their percentage and ranges from 0.34% in NvsV1 to 0.48% in K2vsS comparisons (Fig 4D). Similarly, low impact variants ranges from a 13.71% in LvsS comparison to a maximum of 17.0% in NvsV1 comparison. Likewise, the moderate category of impacts were in range of 12.19% to 16.84% in K2vsS and NvsV1 pair wise

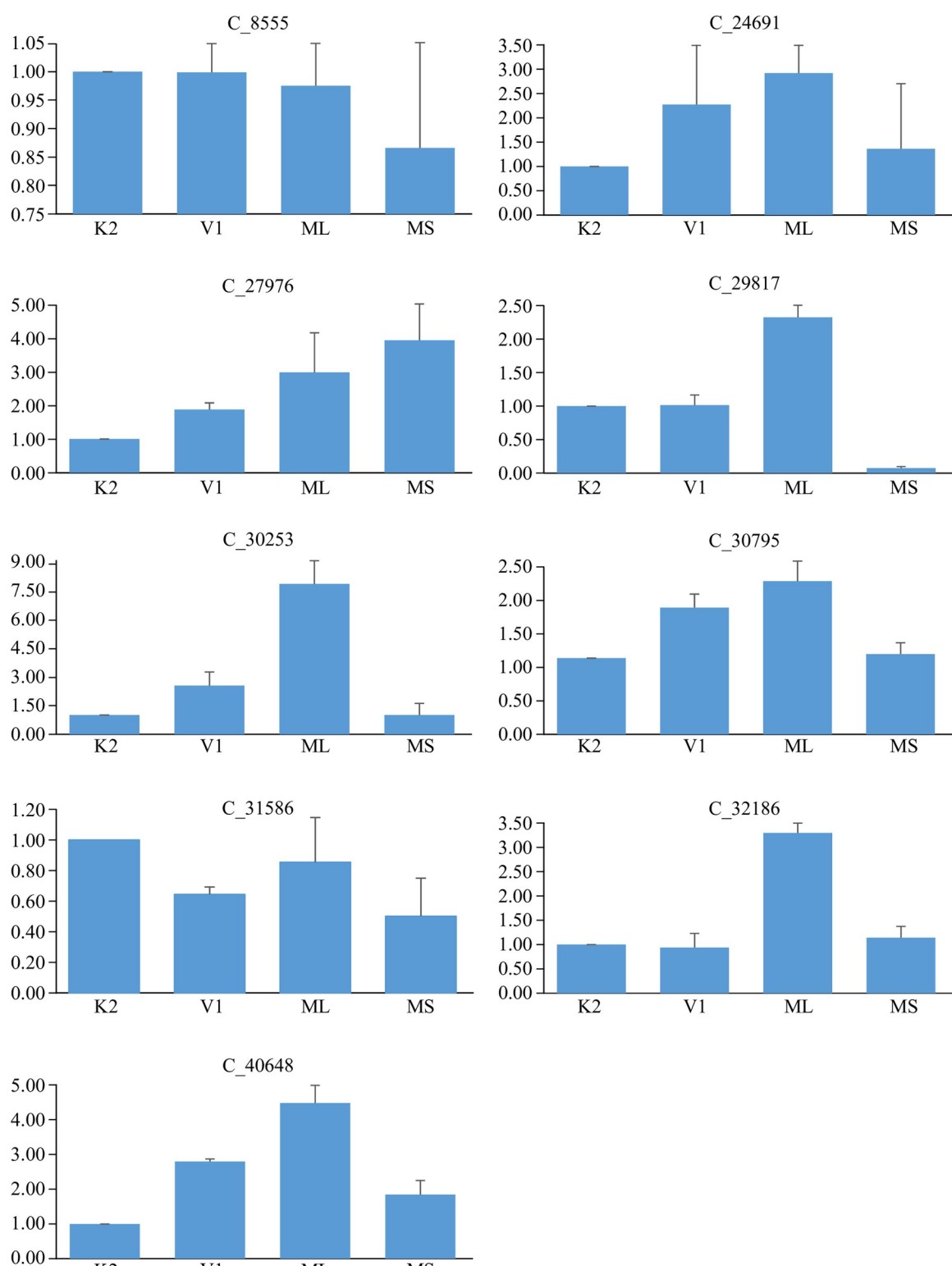

**Fig 3. Bar graphs showing the relative expression pattern obtained in leaf samples of five genotypes studies.δδCt method was applied to calculate the relative fold changes.** X axis represents the fold change and Y axis represents the stages. Error bars represent the standard errors. Person correlation coefficient was also calculated for data obtained from RNA-seq and qPCR (See text for details).

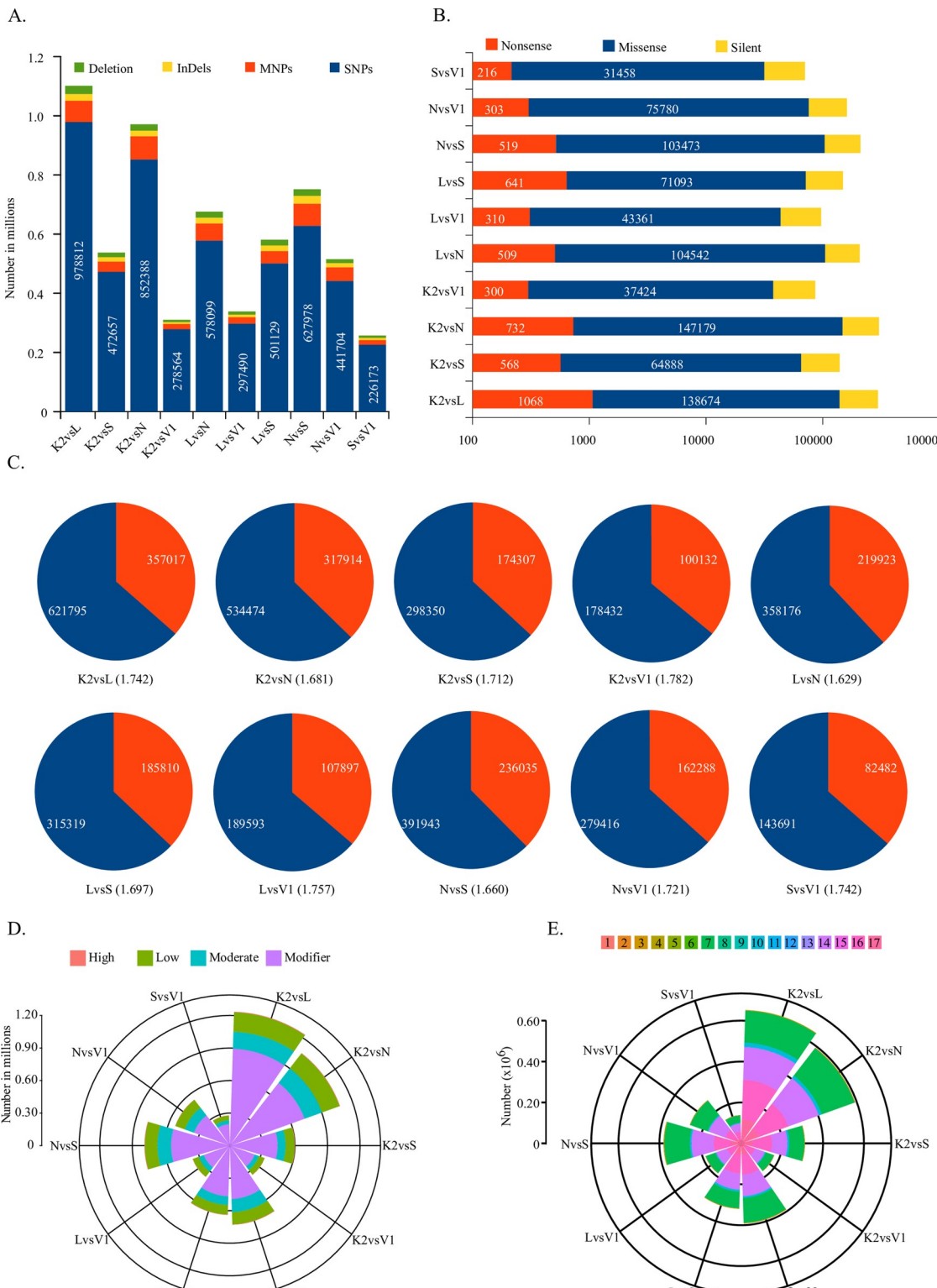

**Fig 4. Figure showing the results of variant finding and their effect on transcriptome.** A) Bar graph showing the number of variants from different categories i.e. SNP, MNP, Insertions and deletions. B) Horizontal bars showing the number of SNPs having effects like non-sense, miss-sense and silent mutations on a log scale. C) Pie charts showing the SNPs type into transition (blue shade) and transversion events (red shade). D) Polar graph with stacked columns showing distribution of SNPs into their strength of their effect on coding sequences. E) Polar stacked columns showing the Effect of variants. Coloured legends details are as 1. Codon Change,

2. Codon Change Plus Codon Deletion, 3. Codon Change Plus Codon Insertion, 4. Codon Deletion, 5. Codon Insertion, 6. Frame Shift, 7. Non Synonymous Coding, 8. Non Synonymous Start, 9. Non Synonymous Stop, 10. Start Gained, 11. Start Lost, 12. Stop Gained, 13. Stop Lost, 14. Synonymous Coding, 15. Synonymous Stop, 16. UTR 3' and 17. UTR 5'.

comparisons respectively. A conclusive analyses of these variants were performed for their impact on different genomic locations like CDS, 5'UTR, 3'UTR etc. A total of 17 such categories were identified in which these variants fall (Fig 4E).

## Ontology enrichment analysis of unique gene: A potential clue about the way they shape the genotypes

Gene ontology enrichment analyses offer a more robust method to assess the functions and other such information about genes. Gene ontology database is rich, could be used for a deeper understand the role of proteins. To find out the role played by those genes that have no homologs in other genotypes, their ontology enrichment was done. Top 200 genes (by FPKM values) were selected to make the data more represent able. In K2 biological process gene ontology category genes with variations showed significant (hypergeometric distribution test; p val$\leq 0.05$ with BenaminiHocherberg correction) enrichment of terms like ion transport, hydrogen transport, proton transport, monovalent inorganic cation, transport, cation transport, chitin metabolic process, starch metabolic process, aminoglycan metabolic process, amine metabolic process, cellular polysaccharide, biosynthetic process, cellular macromolecule and biosynthetic process (Fig 5). Similarly when the unique genes of ML were assessed, the significantly enriched GO biological process category include GO terms like small GTPase mediated signal, transduction, intracellular signal transduction, intracellular signaling pathway, regulation of cellular process, signal transduction, signaling pathway, regulation of biological process, signal transmission, signaling, signaling process, biological regulation, biological_process and cellular process (S1 Fig in S1 File). No significant biological process GO terms could be found from MN and MS genes that harbour variations. From V1 samples the significantly enriched GO terms from biological process include cellular metabolic process, small molecule metabolic process, lipid metabolic process, lipid biosynthetic process, cellular lipid metabolic process, cellular ketone metabolic process, cellular biosynthetic process, organic acid metabolic process, small molecule biosynthetic process, organic acid biosynthetic process, biosynthetic process, cellular protein metabolic process, cellular macromolecule metabolic process and primary metabolic process (S2 Fig in S1 File). These analyses indicate that the genes that are unique to the genotypes could be regulating the above mentioned biological processes that culminate into the unique features shown by these genotypes.

Similarly, when the same analyses were extended with cellular components GO terms, in K2 genotype chloroplast stroma, chloroplast, amyloplast, plastid stroma, chloroplast thylakoid membrane, chloroplast part, chloroplast envelope, chloroplast thylakoid, plastid and plastid thylakoid membrane terms were significantly enriched (S3 Fig in S1 File). While assessing the cellular components GO terms, in ML genotype the significantly enriched terms were heterochromatin, chromatin, large ribosomal subunit, chromosome, non-membrane-bounded organelle, intracellular, non-membrane-bounded organelle, nucleosome, chromosomal part and ribosome (S4 Fig in S1 File). In the same analysis for MN genotype, chromatin, nucleosome, nuclear lumen, protein-DNA complex, chromosomal part, chloroplast inner membrane, intracellular organelle lumen, plastid inner membrane, nuclear part and protein complexcellular component GO terms were enriched (S5 Fig in S1 File). In MS genotype, the cellular component GO terms of unique gene were enriched in intracellular part, cytoplasm, intracellular, non-membrane-bounded organelle, intracellular organelle, non-membrane-

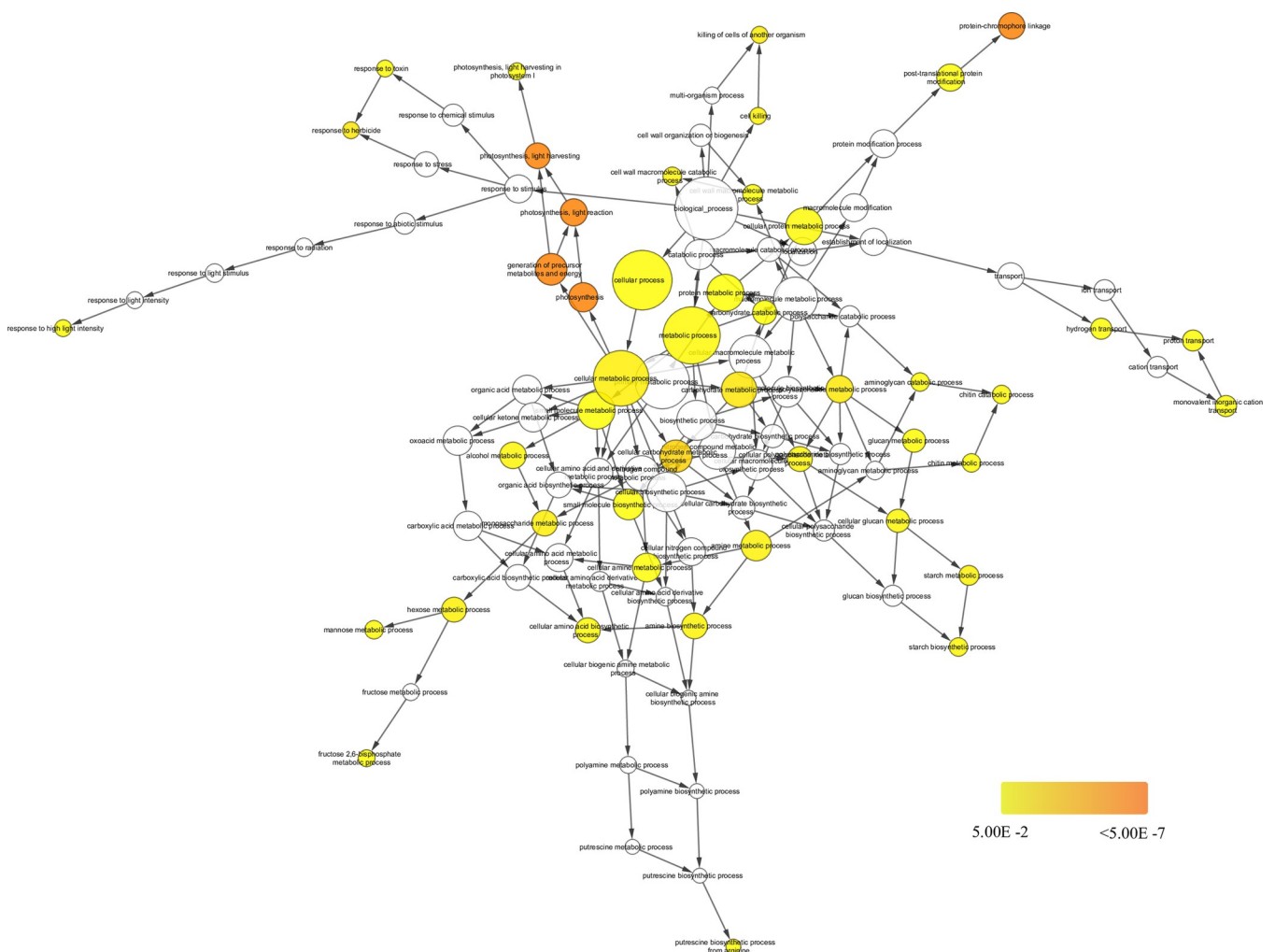

**Fig 5. Network of GO terms of biological process showing the enriched terms for the unique genes of K2 (MI) genotype.** Nodes represent the GO terms. Size of the nodes represents the number of gene and the colour represent significant value (See scale).

bounded organelle, cell part, intracellular, cellular_component and organelle (S6 Fig in S1 File). While running the same analyses for V1 genotype, the unique genes to the genotype have shown significant enrichment in mitochondrion, chloroplast part, cytoplasm, plastoglobule, cytoplasmic part, intracellular organelle part, cytosol, plastid part, small ribosomal subunit and large ribosomal subunit cellular component category (S7 Fig in S1 File).

The third term of the GO category is molecular function. Molecular function GO terms describe about the activity conducted or regulated by that particular gene. While performing molecular function GO terms enrichment analyses in K2 genotype for the genes that have shown exclusive expression, GTPase activity, nucleoside-triphosphatase activity, inorganic diphosphatase activity, pyrophosphatase activity, pyrophosphatase activity, hydrogen-translocating, phosphorus-containing anhydrides, hydrolase activity, acting on acid, transmembrane transporter, P-P-bond-hydrolysis-driven, 5-methyltetrahydropteroyltriglutamate-homocysteine, anhydridesactivity, S-methyltransferase activity terms were found to significantly enriched (S8 Fig in S1 File). Similarly, the genes that are unique to ML tissue shiwed significant enrichment in nucleoside-triphosphatase activity, GTPase activity, pyrophosphatase activity,

phosphorus-containing anhydrides, hydrolase activity, acting on acid, hydrolase activity, acting on acid anhydrides, anhydrides, cytochrome as acceptor, oxidoreductase activity, acting on diphenols and related substances, ubiquinol-cytochrome-c reductase, oxidoreductase activity and S-methyltransferase activity in this category (S9 Fig in S1 File). When this analysis was extended to MN genotype, the significantly enriched terms for this category were carbon-carbon lyase activity, fructose-bisphosphate aldolase activity, aldehyde-lyase activity, GTPase activity, nucleoside-triphosphatase activity, pyrophosphatase activity, hydrolase activity, acting on acid, ammonia ligase activity, acid-ammonia (or amide) ligase activity, glutamate-ammonia ligase activity and hydrolase activity acting on acid anhydrides (S10 Fig in S1 File). In this category no significant GO terms were reported from MS genotype. Further, when this analysis was extended to V1 tissue, terms like ion binding, substrate-specific transporter activity, transmembrane transporter activity, cation binding, pigment binding, transporter activity, hydrolase activity, catalytic activity, ligase activity, forming, nitrogen-metal bonds, forming coordination complexes, ligase activity, forming nitrogen-metal bonds and hydrolase activity, acting on acid anhydrides were showed to be enriched significantly(S11 Fig in S1 File).When binary cut clustering method was applied [30], In K2 biological process category Amine and carbohydrate metabolism related GO terms were highlighted (Fig 6A). Similarly, in K2 molecular function GO terms dehydrogenase was highlighted. Cytoskeleton, Energy metabolism, Intracellular and non-membrane bound organelle, biosynthesis related GO terms were highlighted in K CC, ML BP, MN CC and V1 BP binary cut clusters. Further, in V1 MF and V1 CC category Oxidoreductase and Cytosolic, Ribosomal and Mitochondria related GO terms were found to be highlighted these analyses. This suggest that these genotype vary with respect to these GO terms and may be regulating the genotypes charactersticgworth and development.

In summary these analysis found that the genes that are unique to the genotypes control and regulates various biological processes, their localization are different in the cells and they perform different molecular function that in turn shape the genotype specific traits.

## Discussion

In the present study, we have sequenced and analyzedthe transcriptome of leaves of four Indian genotypes including *M. indica* variety K2 and V1, *M. laevigata* and *M. serrata*. These sequences along with a Chinese haploid variety *M. notabilis*were assembled to gain insights into the constituents of leaves of different species/varieties at transcriptome level. A recent detailed comparative study of 99 transcriptome assemblies, generated with six *de novo* assemblers viz. CLC genomics, SOAPdenovo, Oases, TransAByss, Trinity and NextGEN-esuggested that Trinity is one of the best assemblers amongst the in producing superior transcriptome assembly on the basis of accuracy of unigenes, N50 values and proportion of mappable reads [31]. Another study was conducted involving comparisons of *de novo* transcriptomes assemblers in diploid and polyploid species using peanut (*Arachisspp*) RNA-seq data. In this study transcriptome assemblies were generated for diploid and tetraploid species of *Arachis* by using three different assemblers namely SOAPdenovo-Trans, TransAByss and Trinity in view of finding a better assembler for polyploids [32]. On detailed analysis of the assemblies constructed using these three assemblers, Trinity was found to have a high success rate in sequence assembly in comparison to SOAPdenovo and TransAByss. Additionally, it was also deduced that for diploids, Trinity and TransAByss performed better while in case of complex tetraploids, Trinity performed better [32]. Mulberry leaves used as feed for silkworm are a typical example of plant-herbivore interaction [33] and this interaction since ages has been well studied in plant defence-insect adaptation [34, 35].

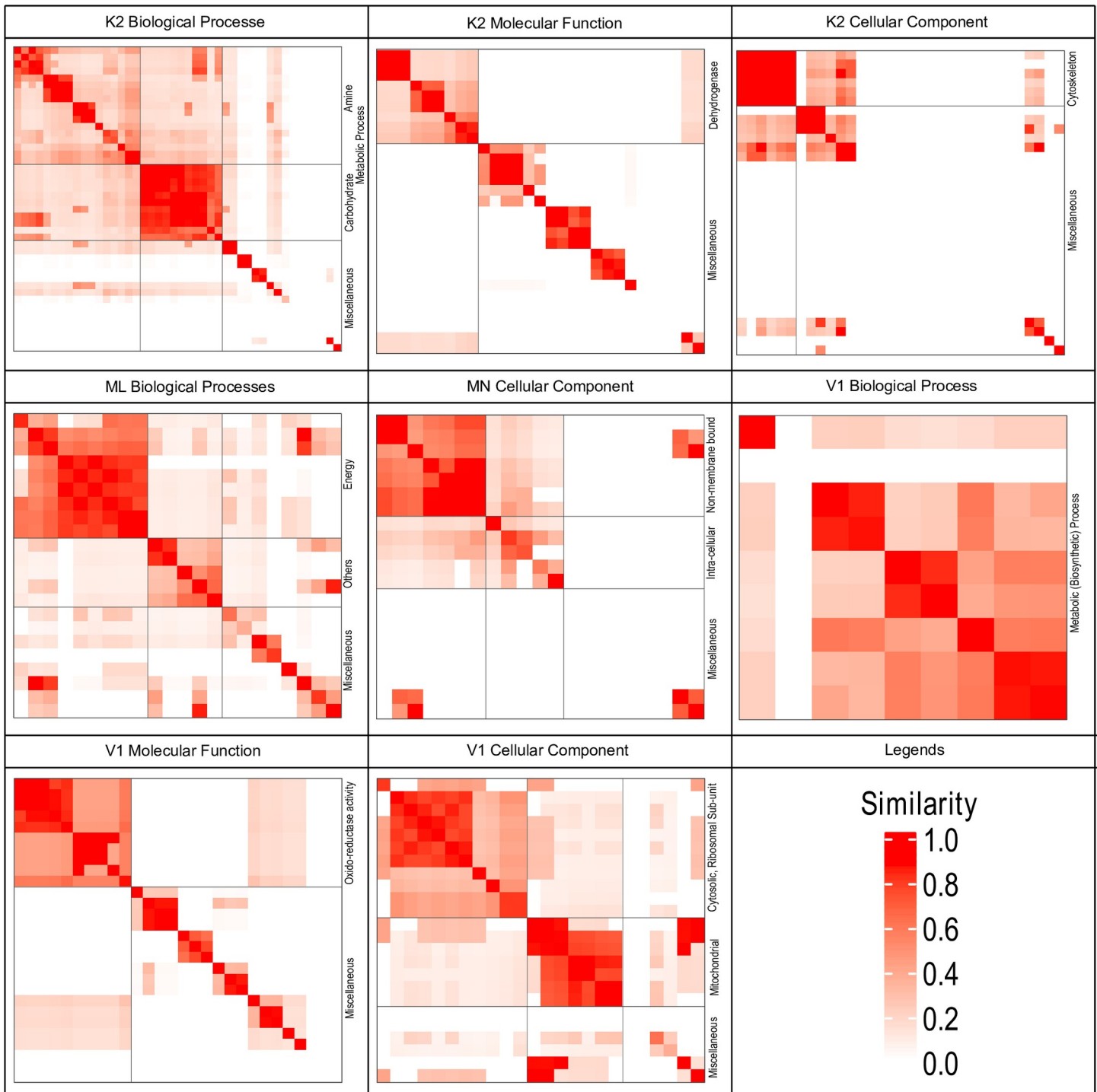

**Fig 6. Similarity matrices of unique GO terms obtained for K2, V1 (*Morus indica* cultivars) and ML, MN and MS leaf transcripts.** Similarity matrices are represented using heat maps that represent the similarity among GO terms and cluster them using binary cluster method. (See legend for similarity levels).

Cytochrome p450 are the defence molecules produced by plants involved in the biosynthesis of secondary metabolites for preventing herbivory. Recently, 174 members of this gene family have been identified in MN, divided into nine clans and 47 families found to be

expressed in tissue specific manner [36]. The expression profile of this gene in our study indicates its higher expression in MI varieties K2 and V1 while lower expression in ML and MS according to differential expression as depicted by heat map and qPCR analysis. *Morus indica* leaves being rich source of nutrition for silkworm has always been preferred for silkworm rearing. AP2, ethylene like responsive transcription factor are the family of genes important for growth and development of plants and are also involved in biotic and abiotic stress. 145 members of this family have been found in *M. notabilis* [37] and detailed analysis of two genes of this family inferred that they are involved in flooding stress [37]. UGTs are the genes involved in detoxification and excretion of both endogenous and exogenous compounds in insects. 42 members of this family have been identified in *Bombyx mori* involved in physiological functions in the insect [38]. Ubiquitin proteasome degradation pathway is carried out by ubiquitin, a small highly conserved gene which forms conjugates with its substrates and mediates the degradation pathway. *Bombyx mori* cytoplasmic polyhedrosis virus (BmCPV), is a pathogen of silkworm causing losses to silkworm industry. One of the ubiquitin like gene from *B. mori* has been identified which may be involved in CPV infection hence providing a cue that silkworm can be benefited from mulberry repertoire of ubiquitin-conjugating enzyme E2 [39]. Aquaporins belong to major intrinsic family (MIPs) proteins in plants implicated in the regulation of cellular transport of water and some molecules. Expression of certain aquaporins have been reported to play critical roles in carbon assimilation and hydraulic dynamics during drought stress and recovery pointing towards their role in drought stress, which is one of the major abiotic stress affecting yield of mulberry [40]. Taken together, we have found that genes including Cytochrome P450, AP2/ERFs, UGTs, Aquaporins etc that are promoting the feed preferences are expressed in lower amount in wild types and higher in cultivated varieties.

Previously, a total of thirty-six MIPs have been reported in *M. notabilis* genome [5] and their gene ontology analysis have indicated their roles in plant development and stress management. In a study Short-chain dehydrogenase/reductase family member was found to be involved in metamorphosis of larvae [41]. Zinc fingers have been associated with variety of stresses in many species of plants especially abiotic stress. They are abundant in mulberry too. In a study, on finding their expression pattern most of the zinc finger proteins analyzedwas found to be stress-responsive. Zinc finger CCCH-type family protein were strongly induced in leaves [42]. In the present analysis, a plethora of genes were investigated pertaining to the transcriptomes of ML, MS, MI (K2 and V1) and MN whose expression was differentially regulated. Selection of some of the genes for functional validation by qRT-PCR affirms the differential regulation provided by the *in silico* analysis. All of the genes selected for validation, stipulates towards the functional relevance and significance of these genes for the varietal improvement, with respect to mulberry-silkworm association and stress management.

Variant identification has been a favoured approach for species improvement now a day. SNP mining and validation for finding out intraspecific variation in response to drought stress is being used for crop enhancement [43]. Association genetics posing problems for marker identification leads to a better approach which is transcriptome based identification of markers and variants especially for polyploid species and hence transcriptome based search for markers and variants, can be used effectively for polyploid species of mulberry as it exhibits varied ploidy levels amongst the germplasm [44]. Next generation sequencing technologies can be utilized for resequencing entire genomes or transcriptomes effectively of related genomes to elucidate the genetic diversity among species or germplasm for crop improvement [45].

## Conclusions

With the advent of Next Generation Sequencing techniques (NGS), difference between model and non model systems are being done away. *Morus*, being a non model crop pose the challenges such as diversity, non standardized protocols and unavailability of a chromosomal level assembly. However, a larger chunk of population in India depends upon *Morus* cultivation and silk worm rearing for their daily wages. We have sequenced the transcriptome of five genotypes assembled them which could have huge utilities in *Morus* breeding programs. Out of these five genotypes, two are cultivated including K2 and V1 which provide a ready reference to compare with wild varieties. While assembling the transcriptomes, Trinity software was found to perform best for *Morus*. In this analysis, for the production of superior transcriptome assembly different assemblers widely used in NGS analysis, were studied in which Trinity was found to be the optimally suited assembler. Further, transcriptome of leaves of the genotypes in the study depicted large number of differentially expressed transcripts. Contrasting expression profiles of all thetranscriptome of leaves incorporated in the study revealed the presence of essential genes regulating stress responsive pathways and plant defence-insect adaptation.Upon assembly, an experiment was performed to identify unique genes and their 'nische' to get an idea about how they shape the genotypes. Various unique genes showed GO term enrichment for different biological process, molecular function and cellular components. In depth exploration of the unique transcripts and their gene ontology enrichment reveals their association chiefly with photosynthesis, carbohydrate metabolism, protein turnover, pathogenesis and signallingfurther directing towards the significance of these genes for varietal improvement in association with mulberry-silkworm interaction and stress mechanisms. We have also found that several feed behaviour promoting genes including AP2/ERF, UGT, Cytochrome P450 and Aquaporins are modulated in cultivated varieties which render them more palatable. These results indicate the acquisition of genotype specific genes which might shape the morphology, anatomy and other such physiological factors helping in genotype specific traits. Transcriptome wide variant analyses were also performed to find out the variants among these five genotypes. This data could be a valuable asset for *Morus* breeding community where genomic improvement programs are being carried out.

## Methods with statistics section

### Plant material

To isolate the total RNA for transcriptome sequencing, fully grown mature leaves were taken from two wild type and two cultivated varieties. Two wild types include *Morus laevigata* (ML) and *Morus serrata* (MS) and two cultivated varieties including K2 and V1 were taken. Mature leaves samples of ML were harvested from a fully grown tree maintained in the Botanical Garden of Department of Botany, University of Delhi, Delhi. Similarly, mature leaves of MS, K2 and V1 were harvested from a fully grown tree maintained in Department of Plant Molecular Biology, University of Delhi South Campus, Delhi. Immediately after harvesting the leaf samples, they were snap chilled in Liquid nitrogen and stored in -80˚C deep freezer for later usage.

### Total RNA isolation and quality check

Total RNA were isolated from approximately 100 mg of the leaf tissue from each sample. For ML and MS details are already published NCBI SRA accession numbers SRX1515878 and SRX1506562 respectively[21]. Modified GITC method was used to isolate total RNA samples. Isolated RNA was purified on silica based columns provided in Qiagen Plant Mini Kit (Qiagen, USA). The samples were treated with DNAse in order to remove DNA contamination. These

purified samples were subjected to quantitative and qualitative checks. Quantitative estimations were done on NanoVue (GE Healthcare). Integrities of the samples were checked on Agilent 2100 Bioanalyzer® with RNA Nano Kit before proceeding to library preparations. Aliquots of same RNA samples were kept aside for qPCR experiments.

## Library preparations and sequencing of the transcriptomes

QC passed samples were used for library preparation. Preparation of library was done with Illumina provided TruSeq RNA Sample preparation kit and the manufacturer's instructions were followed in strict sense. After the preparation of library, samples were sequenced on Illumina HiSeq2000 platform by a commercial sequencing service provider (Scigenome Labs Pvt. Ltd, Cochin, Kerala, India). In brief, post RNA enrichment using oligo dT primers, second strand synthesis was performed. After the second strand synthesis, the generated cDNA samples were fragmented and purified as per the given protocols. Post fragmentation, ligation was done with provided adapters. Post ligation samples were barcoded and sequenced.

## Pre assembly cleaning, assembly, annotation and identification of homologs

Trimmomatic version 0.30 [46], was used for removal of the adapter sequences, primer sequences and their fragments along with bad reads, with the default settings of LEADING:5, TRAILING:5 and MINLEN:36. To identify the better assembler, five *de novo* assembler were used including CLC Genomics, Trinity [27], SoapDeNovo [26] and TransAByss [25]. ML reads were used as default sample with default parameters of every program. Highest numbers of raw sequences were there in the ML sample thus it was used with for all the assemblers to identify the most compatible one. The criteria to select default assembler for further analyses were resultant with higher number of full length transcripts and higher N50 values. In this analysis, Trinity faired with respect to other assemblers used therefore used as default assembler in later course. To assemble the other genotypes' transcriptomes, cleaned paired fastq files with default parameters of Trinity were used. In silico normalization was done with default parameters given as in Trinity.

To annotate assembled transcriptomes, Annocript pipeline was used [28]. Annocript uses BLASTx,UniRef database, SwissProt databases, Rfam, Conserved Domains Database (CDD) of NCBI and Pfam database to annotate the sequences. It also identifies complete and incomplete CDS from the assembled transcriptome and also assigns gene ontology accession based on similarities with known annotated sequences.

Homolog identification was done in order to identify common and unique genes from the assembled transcriptomes. To identify the homologs among the assembled transcriptomes, ProteinOrtho [29] was used. In downstream analyses, only the homologs common to all the species were used. Orthologyassesment among the five species of *Morus* has been performed using it. Proteinortho implements an extended reciprocal blast hit method to identify orthologous clusters between or among the compared species. In our analysis, we have made ten pairwise analyses to find orthologous clusters in all permutations and combinations.

## Identification of genes unique to the genotypes and their gene ontology enrichment

Orthologs in five genotypes of the mulberry have been generated using ProteinOrtho. The resultant thus obtains were in form of clusters with high algebraic connectivity. These clusters have genes from genotypes that are othologs. To identify unique genes from particular

genotypes, such clusters were chosen which harbors genes only from a single genotype and their orthologs were not reported. From these cluster lists for each genotype unique genes were prepared and used for downstream analyses. During the annotation process, the program Annocript, also gives the output for GO categories. Thus GO terms for these uniques genes were fetched from the annotation obtained post Annocript run. For GO enrichment analyses, top 200 genes from each genotype (high TPM values)were taken and fed in Bingo plugin [47] in Cytoscape version 3.02 [48]. GO terms for these genes were used with the GO terms of whole transcript set was used as reference and hyper-geometric distribution test with Benjami-niHocherberg's correction was applied. Only the significantly enriched terms (p < = 0.05) were shown and kept for the interpretations.

## Abundance estimation, differential expression analayses and qPCR based validation of gene expression

Post assembly and after homologs identification, a custom reference file of homolog sequences were prepared. This custom transcriptome file of homolgs was used as reference to align reads from different genotypes. A perl script align_and_estimate_abundance.pl, which is based on bowtie2 [49], provided in the Trinity software was used for the same purpose. Post abundance estimation, edgeR package [50] of R was used to estimate FPKM values and calculate differential expression profile. PtR script of Trinity was used to make heatmap of differentially expressed genes for visual representation.

## SSRs, indels, SNP and other variants identification

To identify simple sequence repeats (SSRs) perl script obtained from Gramene was used [51]. The script was modified with the number of repetitive units for different types of repeats as Di-9, Tri-6, Tetra-5, Penta-5 and Hexa-4. Rest including hepta and more were with three repeats. To identify the variants, assembled transcriptome were used as a reference for aligning reads in possible 10 binary combinations (one genotype vs. other). Only high quality filtered reads were aligned which have passed the quality criteria mentioned earlier. The alignment was done using Bowtie2 version 2.2.4 with default parameters. BAM alignment files generated in the previous step were used as an input for FreeBayes version 0.9.20-8-gfef284a [52] to call various variants from all ten combinations. Further, the resultants were filtered using vcftools version 0.1.13 [53], using stringent parameter including quality score of at > = 30, variant support by a minimum read depth of 10 a frequency of 100%. Two or more in-dels were also filtered if they fall in a 10 bp window and SNPs within three bp of an InDel. Finally the filtered variants were annotated for their effect on transcript using SNPeff [54] and different aspects like their transition and transversion ratio, Effect on coding properties, genomic regions etc were estimated.

## Estimation of transcript abundance using qPCR

**Same total RNA samples were used to estimate the transcript abundance using qPCR technique.** The qualitative and quantitative estimation of samples were performed again and were checked on spectrophotometer and reducing agarose gel electrophoresis. 1.0 microgra-mof total RNA sample was used to generate cDNA using RevertAid First Strand CDNA synthesis kit (Thermo Scientific)in a 20 μl reaction setup in a thermocycler. Primers were designed in Primer Express software using the sequences obtained after assembly. Their annotation and uniqueness was confirmed by performing BLASTn search againstthe transcript database of MN. PCR was performed in StratageneMx3005P (Agilent Technologies) with two biological and three technical replicates with default cycles. Brilliant II SYBR Green qPCR

Master Mix (Agilent Technologies) was used to set up the reaction for the PCR cycle with 1 μl of cDNA. Ct values obtained after running the PCR protocol was used. Actin was used as an internal control and its Ct value was used to normalize the data. Delta delta Ct method was used to find out the fold changes exhibited by these transcripts in different genotypes. Heteroscedastic Student's t-test was used in pairwise comparison with K2 delta Ct values to find out the significance level.

## Data access

The sequence data for leaf transcriptome of *Morus laevigata* and *Morus serrata* is already submitted to NCBI SRA database with accession numbers SRP068061 and SRP067869 respectively by our lab previously [23]. The read data for Morus notabilis leaf transcriptome was downloaded from NCBI SRA database with accession number SRA075563. The sequence data for *Morus indica* cv K2 and cv V1 could be accessed under the Bioprojects accessions PRJNA717975 and PRJNA717991 respectively.

## Supporting information

**S1 Table. Showing the assembly result obtained from different assemblers used when *M laevigata* reads were utilized to find out the better assembler.**
(XLS)

**S2 Table. Showing the annotation of the genes selected for qPCR based validation of expression profiles exhibited by NGS data analyses.**
(XLS)

**S3 Table. Showing the relative fold changes obtained from qPCR and RNAseq.** Pearson's coorelation coefficient was calculated for these two data sets and given in the separate column.
(XLSX)

**S1 File.**
(PDF)

## Acknowledgments

VKB and PK are thankful to the Department of Biotechnology, Ministry of Science and Technology, Government of India for providing grants for the current research work.

## Author Contributions

**Conceptualization:** Paramjit Khurana.

**Data curation:** Vinay Kumar Baranwal.

**Formal analysis:** Vinay Kumar Baranwal.

**Investigation:** Vinay Kumar Baranwal, Nisha Negi.

**Methodology:** Vinay Kumar Baranwal, Nisha Negi.

**Project administration:** Vinay Kumar Baranwal.

**Supervision:** Paramjit Khurana.

**Writing – original draft:** Vinay Kumar Baranwal, Nisha Negi.

**Writing – review & editing:** Paramjit Khurana.

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
