## [Decision Letter · Decision Letter 0]

1 Mar 2021

PONE-D-21-02525

Comparative transcriptomics of leaves of five mulberry accessions and cataloguing structural and expression variants for future prospects

PLOS ONE

Dear Dr. Khurana,

Thank you for submitting your manuscript to PLOS ONE. After careful consideration, we feel that it has merit but does not fully meet PLOS ONE’s publication criteria as it currently stands. Therefore, we invite you to submit a revised version of the manuscript that addresses the points raised during the review process.

Two reviews have given constructive comments for improvement of the MS. It is requested to kindly revise the manuscript considering the comments of the Reviewers.

We look forward to receiving your revised manuscript.

Kind regards,

Mukesh Jain

Academic Editor

PLOS ONE

Journal Requirements:

Reviewers' comments:

Reviewer's Responses to Questions

**Comments to the Author**

1. Is the manuscript technically sound, and do the data support the conclusions?

Reviewer #1: Yes

Reviewer #2: Yes

2. Has the statistical analysis been performed appropriately and rigorously? 

Reviewer #1: Yes

Reviewer #2: Yes

3. Have the authors made all data underlying the findings in their manuscript fully available?

Reviewer #1: No

Reviewer #2: Yes

4. Is the manuscript presented in an intelligible fashion and written in standard English?

Reviewer #1: Yes

Reviewer #2: Yes

5. Review Comments to the Author

Reviewer #1: Title: Comparative transcriptomics of leaves of five mulberry accessions and cataloguing structural and expression variants for future prospects

In this manuscript the authors have sequenced the leaf tissues from five species of Morus sp. namely M. indica variety K2 and V1, M. laevigata and M. serrata. and M. notabilis. Initially they have compared the various assemblers used for transcriptome assembly. They have shown that Trinity assembler performs better than the rest of the assemblers. The authors have identified common ortholog transcripts among the five species and seen their differential expression. They have also validated some genes using qRTPCR.

Further the authors have studied the SNP variations in the five genotypes and identified several. The authors have also done the GO enrichment analysis in the five genotypes.

Here are a few comments:

1. In figure 2 the authors should include the description of the contigs used for making the heat map.

2. A comparative heat map for the GO enrichment analysis can be made for all the five genotypes.

3. What are the pathways that are differentially regulated in these five species?

4. Genes that could be playing important role in plant-insect interaction could be selected and SNP identified from those could be highlighted.

5. I could not find any SRA number in the manuscript so I presume the authors have not deposited their data in the public repositories.

Reviewer #2: The article reports interesting analyses where the authors have compared the transcriptome data of five genotypes of mulberry to find out the differentially expressed genes and variants. Here, authors have also tried to find out the impact of variants on different traits that are shown by the genotypes. This study might have larger implications in various mulberry breeding programs and would be a data source for breeding experiments. Besides the scientific significance of the manuscript or the amount of data it encompasses, the manuscript requires some reworking to ensure technical soundness, clarity and readability. My specific comments on this manuscript are as follows;

Abstract: It is not well written, unnecessary extension of several experiments only. The abstract should reflect the actual outcome in a consolidated manner. It is very difficult what the comparisons mean and what they are referring to. Name of cultivars and comparisons do not match.

Major comments:

Introduction should be arranged in a more systematic manner. Research of the several recent workers specifically on the genotypes properties should be cited at relevant places. Authors should also take care of grammatical errors. Therefore, a serious rework to improve this section is advised. Authors should also discuss potential mechanisms that might have shaped these genotypes.

Authors have taken three wild types and two cultivated varieties which show contrasting characters. Although this data is given in the text, it would be better it is given in tabular format along with some characteristic feature of cultivars chosen such as stress tolerance/sensitive. Also, total read number, unigenes, transcripts can be mentioned in the same table

Methods: Here also scope exists to give a structured outlook of this section. Language and grammatical errors should also be corrected by the authors. My specific comments about this sections are given below that must be addressed.

1. Authors have used standalone version of Annocript to annotate the assembled transcriptomes which rely on several databases. They should mention the version of these databases too along with the version of the annocript tool.

2. Authors have used Proteinortho for identification of orthologs. The standalone version of this software is based on finding the best reciprocal BLAST hits. Since similarity based searches are often debated, the authors must provide the details of this step like nature of input sequences, cutoff values, e-values, version of BLAST program etc.

3. Explain algebraic connectivity in little more detail

4. Authors have tried to validate their transcriptome by means of qPCR experiments. It is advisable to give correlation between them by using a coefficient.

5. The results obtained from Gene Ontology enrichment analyses could be extrapolated for genotype specific traits. Although authors have put a section on the same, it is not reflected in the discussion portion.

6. Will authors explain what they mean by GO enrichment analyses of ‘Unique Genes’?

7. It is of utmost importance to deposit the raw sequence files, assembly to a public database for easy accessibility.

8. in the heat map and qpcr graphs, for better understanding, give gene annotation as well, in the present version only the an identity number is given.

9. Important and interesting information is given in supplementary figures related to GO analysis. Can authors present this data in main figure(s) in a way to compare varietal differences in three main figures representing cellular component, molecular function and biological process.

In the discussion section

Discussion should be based on the results that authors have produced rather than presenting their data in this section.

Conclusion should be elaborated more.

English issues:

There are many mistakes throughout the text. Authors must ensure the homogeneity, correctness across the manuscript. A few of them are given below.

1. ... the leaf transcriptomes  transcriptome of leaves.

2....comparison b/w genotypes like KvsV should be uniform everywhere

5. as the data -- as it

Upregulation -- up-regulation throughout the text

Misssense -- miss-sense

de novo - de novo

transabyss - Transabyss throughout the text

Nomenclature of species is inconsistent, some times in italics where as some times in regular

6. PLOS authors have the option to publish the peer review history of their article (what does this mean?). If published, this will include your full peer review and any attached files.

Reviewer #1: No

Reviewer #2: No

---

## [Author Response · Author response to Decision Letter 0]

14 Apr 2021

Reviewer #1: Title: Comparative transcriptomics of leaves of five mulberry accessions and cataloguing structural and expression variants for future prospects

In this manuscript the authors have sequenced the leaf tissues from five species of Morus sp. namely M. indica variety K2 and V1, M. laevigata and M. serrata. and M. notabilis. Initially they have compared the various assemblers used for transcriptome assembly. They have shown that Trinity assembler performs better than the rest of the assemblers. The authors have identified common ortholog transcripts among the five species and seen their differential expression. They have also validated some genes using qRTPCR.

Further the authors have studied the SNP variations in the five genotypes and identified several. The authors have also done the GO enrichment analysis in the five genotypes.

Here are a few comments:

1. In figure 2 the authors should include the description of the contigs used for making the heat map.

Answer: Authors are thankful for the tendered suggestion. As suggested, we have incorporated the putative annotation of the contigs in figure 2.

2. A comparative heat map for the GO enrichment analysis can be made for all the five genotypes.

Answer: Authors are thankful for the suggestion. As asked, we have incorporated a new figure comprising the heat map of unique GO terms. We have used simplifyEnrichment package of R to conduct this analysis. Texts at the relevant places are incorporated in the main manuscript.

3. What are the pathways that are differentially regulated in these five species?

Answer: During the annotation process, we have encountered very few genes that were mapped to the pathways. This is why; the authors have relied on gene ontology terms for their in-depth annotations.

4. Genes that could be playing important role in plant-insect interaction could be selected and SNP identified from those could be highlighted.

Answer: Authors politely decline the tendered suggestion as it is beyond the scope of study that we have hypothesized.

5. I could not find any SRA number in the manuscript so I presume the authors have not deposited their data in the public repositories.

Answer: Authors are thankful for this suggestion. The data have been deposited in the NCBI SRA database with the Bioproject accession number PRJNA717975 and PRJNA717991 for K2 and V1 leaf samples respectively. Data for ML and MS leaf tissues were already submitted to the NCBI Sequence Read Archive (SRP068061 and SPR067869, respectively), which was previously sequenced by our group. The text has been modified at relevant places in the manuscript. 

Reviewer #2: 

The article reports interesting analyses where the authors have compared the transcriptome data of five genotypes of mulberry to find out the differentially expressed genes and variants. Here, authors have also tried to find out the impact of variants on different traits that are shown by the genotypes. This study might have larger implications in various mulberry breeding programs and would be a data source for breeding experiments. Besides the scientific significance of the manuscript or the amount of data it encompasses, the manuscript requires some reworking to ensure technical soundness, clarity and readability. My specific comments on this manuscript are as follows;

Abstract: It is not well written, unnecessary extension of several experiments only. The abstract should reflect the actual outcome in a consolidated manner. It is very difficult what the comparisons mean and what they are referring to. Name of cultivars and comparisons do not match.

Major comments:

Introduction should be arranged in a more systematic manner.

Answer: Suggested changes are accepted and incorporated in the manuscript at the relevant places. The introduction section has been rearranged according to evolutionary trends, economic usages, genotypes traits and the hypothesis of the present work.

Research of the several recent workers specifically on the genotypes properties should be cited at relevant places.

Answer: Suggest changes incorporated at the relevant places. 

Authors should also take care of grammatical errors. Therefore, a serious rework to improve this section is advised.

Answer: Authors are thankful for this suggestion. We have got the revised version of the manuscript proofed by native English speaker.

Authors should also discuss potential mechanisms that might have shaped these genotypes. 

Answer: Authors have received the same advice from the other reviewer. We have incorporated the relevant findings in the discussion section.

Authors have taken three wild types and two cultivated varieties which show contrasting characters. Although this data is given in the text, it would be better it is given in tabular format along with some characteristic feature of cultivars chosen such as stress tolerance/sensitive. Also, total read number, unigenes, transcripts can be mentioned in the same table

Answer: Authors are partially agreeing to the given suggestion. We have incorporated a new supplementary table having details of the phenotypes shown by the selected genotypes in this study. Other details that are asked by the reviewer are already there.

Methods: Here also scope exists to give a structured outlook of this section. Language and grammatical errors should also be corrected by the authors. My specific comments about this sections are given below that must be addressed.

1. Authors have used standalone version of Annocript to annotate the assembled transcriptomes which rely on several databases. They should mention the version of these databases too along with the version of the annocript tool.

Answer: Authors are thankful for detailing out these shortcomings. We have mentioned these details in the manuscript at relevant places. 

2. Authors have used Proteinortho for identification of orthologs. The standalone version of this software is based on finding the best reciprocal BLAST hits. Since similarity based searches are often debated, the authors must provide the details of this step like nature of input sequences, cutoff values, e-values, version of BLAST program etc.

Answer: Proteinortho version 5.16 was used for orthologs detection. Input sequences containing proteins sequences were used for this program. Proteinortho was used with default parameters that includes e value <= 1e-05, minimum percent identity = 25, minimum coverage percent = 50 etc. text in the manuscript is modified at relevant places. 

3. Explain algebraic connectivity in little more detail

Answer: The algebraic connectivity indicates how densely the genes are connected in the orthology graph that was used for clustering. A connectivity of 1 indicates a perfect dense cluster with each gene similar to each other gene.

4. Authors have tried to validate their transcriptome by means of qPCR experiments. It is advisable to give correlation between them by using a coefficient.

Answer: Authors are thankful for the suggestion. Gene-wise Pearson’s correlation coefficient was calculated between the fold change dataset for every gene. Text in the main manuscript was modified at relevant places.

5. The results obtained from Gene Ontology enrichment analyses could be extrapolated for genotype specific traits. Although authors have put a section on the same, it is not reflected in the discussion portion.

Answer: Authors have received the same suggestion from the other reviewer. We have incorporated it in discussion section, as suggested.

6. Will authors explain what they mean by GO enrichment analyses of ‘Unique Genes’?

Answer: By stating some genes as unique, the authors meant that these genes were not expressed in the other genotypes with respect to the selected ones. These genes have reported TPM values as “0” zero in other genotypes. Since, these genes are expressed in a genotypes specific manner, we have refered them as unique genes. After filtering such genes, GO enrichment analyses were performed to know their functions. 

7. It is of utmost importance to deposit the raw sequence files, assembly to a public database for easy accessibility.

Answer: Authors are thankful for this suggestion. The data have been deposited in the NCBI SRA database with the Bioproject accession number PRJNA717975 and PRJNA717991 for K2 and V1 leaf samples respectively. Data for ML and MS leaf tissues were already submitted to the NCBI Sequence Read Archive (SRP068061 and SPR067869, respectively), which was previously sequenced by our group. The text has been modified at relevant places in the manuscript.

8. in the heat map and qpcr graphs, for better understanding, give gene annotation as well, in the present version only the an identity number is given.

Answer: Authors have agreed to the suggestion tendered and same has been incorporated in figure two.

9. Important and interesting information is given in supplementary figures related to GO analysis. Can authors present this data in main figure(s) in a way to compare varietal differences in three main figures representing cellular component, molecular function and biological process?

Answer: Authors politely refuse to follow the tendered advice. The GO enrichment data is represented by several figures and are quite large. We believe that if these figures are incorporated in the main manuscript, it will violate the journals limitation. Also, it is discussed in results section.

In the discussion section

Discussion should be based on the results that authors have produced rather than presenting their data in this section.

Answer: As suggested, we have incorporated several changes in the discussion section. Authors are thankful.

Conclusion should be elaborated more.

Answer: Suggested changes are incorporated in the conclusion section. Authors are thankful.

English issues: (Taken care as per the suggestions)

There are many mistakes throughout the text. Authors must ensure the homogeneity, correctness across the manuscript. A few of them are given below.

1. ... the leaf transcriptomes  transcriptome of leaves.

2....comparison b/w genotypes like KvsV should be uniform everywhere

5. as the data -- as it

Upregulation -- up-regulation throughout the text

Misssense -- miss-sense

de novo - de novo

transabyss - Transabyss throughout the text

Nomenclature of species is inconsistent, some times in italics where as some times in regular

Answer: All the above made suggestion along with other typographical mistake have been corrected. Authors are thankful.

---

## [Decision Letter · Decision Letter 1]

12 May 2021

Comparative transcriptomics of leaves of five mulberry accessions and cataloguing structural and expression variants for future prospects

PONE-D-21-02525R1

Dear Dr. Khurana,

We’re pleased to inform you that your manuscript has been judged scientifically suitable for publication and will be formally accepted for publication once it meets all outstanding technical requirements.

Kind regards,

Mukesh Jain

Academic Editor

PLOS ONE

Additional Editor Comments (optional):

Reviewers' comments:

Reviewer's Responses to Questions

**Comments to the Author**

1. If the authors have adequately addressed your comments raised in a previous round of review and you feel that this manuscript is now acceptable for publication, you may indicate that here to bypass the “Comments to the Author” section, enter your conflict of interest statement in the “Confidential to Editor” section, and submit your "Accept" recommendation.

Reviewer #2: All comments have been addressed

2. Is the manuscript technically sound, and do the data support the conclusions?

Reviewer #2: Yes

3. Has the statistical analysis been performed appropriately and rigorously? 

Reviewer #2: Yes

4. Have the authors made all data underlying the findings in their manuscript fully available?

Reviewer #2: Yes

5. Is the manuscript presented in an intelligible fashion and written in standard English?

Reviewer #2: Yes

6. Review Comments to the Author

Reviewer #2: (No Response)

7. PLOS authors have the option to publish the peer review history of their article (what does this mean?). If published, this will include your full peer review and any attached files.

Reviewer #2: **Yes: **Harsh Chauhan

---

## [Editor Report · Acceptance letter]

5 Jul 2021

PONE-D-21-02525R1 

Comparative transcriptomics of leaves of five mulberry accessions and cataloguing structural and expression variants for future prospects 

Dear Dr. Khurana:

I'm pleased to inform you that your manuscript has been deemed suitable for publication in PLOS ONE. Congratulations! Your manuscript is now with our production department. 

Kind regards, 

on behalf of

Dr. Mukesh Jain 

Academic Editor

PLOS ONE